# NRF2, a Superstar of Ferroptosis

**DOI:** 10.3390/antiox12091739

**Published:** 2023-09-08

**Authors:** Ruihan Yan, Bingyi Lin, Wenwei Jin, Ling Tang, Shuming Hu, Rong Cai

**Affiliations:** Department of Biochemistry & Molecular Cell Biology, Shanghai Jiao Tong University School of Medicine, Shanghai 200025, China; super_han@sjtu.edu.cn (R.Y.); linbingyi@sjtu.edu.cn (B.L.); ref-rain@sjtu.edu.cn (W.J.); tanglingyi@sjtu.edu.cn (L.T.)

**Keywords:** NRF2, ferroptosis, antioxidant, metabolism

## Abstract

Ferroptosis is an iron-dependent and lipid peroxidation-driven cell death cascade, occurring when there is an imbalance of redox homeostasis in the cell. Nuclear factor erythroid 2-related factor 2 (NFE2L2, also known as NRF2) is key for cellular antioxidant responses, which promotes downstream genes transcription by binding to their antioxidant response elements (AREs). Numerous studies suggest that NRF2 assumes an extremely important role in the regulation of ferroptosis, for its various functions in iron, lipid, and amino acid metabolism, and so on. Many pathological states are relevant to ferroptosis. Abnormal suppression of ferroptosis is found in many cases of cancer, promoting their progression and metastasis. While during tissue damages, ferroptosis is recurrently promoted, resulting in a large number of cell deaths and even dysfunctions of the corresponding organs. Therefore, targeting NRF2-related signaling pathways, to induce or inhibit ferroptosis, has become a great potential therapy for combating cancers, as well as preventing neurodegenerative and ischemic diseases. In this review, a brief overview of the research process of ferroptosis over the past decade will be presented. In particular, the mechanisms of ferroptosis and a focus on the regulation of ferroptosis by NRF2 will be discussed. Finally, the review will briefly list some clinical applications of targeting the NRF2 signaling pathway in the treatment of diseases.

## 1. Introduction

Ferroptosis was first observed in 2003 by Dixon, in whose observation, cancer cells with RAS mutations were selectively eliminated by erastin, which was newly discovered by Dolma and his colleague [1]. Subsequently, Yagoda [2] and Yang [3] in 2007 and 2008, respectively, found that iron chelators could inhibit this kind of cell death and discovered another compound, RSL3 (RAS synthetic lethal 3), which could activate it. In 2012, when studying the mechanism behind erastin killing cancer cells, Dixon et al. officially coined this mode of death as ferroptosis based on its unique features [4].

The morphological, biochemical, and genetic characteristics of ferroptosis are distinct from other types of cell death [4]. Morphologically, ferroptosis does not perform typical features of necrosis (such as plasma membrane breakdown and generalized swelling of the cytoplasm and organelles), neither does it have the properties of apoptosis (such as cellular and nuclear volume reduction, nuclear fragmentation, and formation of apoptotic bodies). The main recognizable ultrastructural characteristic of ferroptosis is the apparently altered mitochondrial morphology. Earlier studies showed that mitochondria in ferroptosis cells were significantly contracted along with increased membrane density [2,4]. Later studies illustrated that ferroptosis caused mitochondrial swelling, reduced cristae, and caused outer membrane rupture [5,6,7,8]. Biochemically, there are prominent features that can distinguish ferroptosis from others. For example, intracellular depletion of GSH (glutathione), decreased activity of GPX4 (glutathione peroxidase 4), and accumulation of lipid peroxides are regarded as essential factors to promote ferroptosis [3,5]. In genetics, ferroptosis is a biological process regulated by multiple genes, involving regulatory mechanisms in iron homeostasis, lipid peroxidation metabolism, amino acid metabolism, and so on.

Just as its name implies, NRF2 (nuclear factor erythroid 2-related factor 2, NFE2L2) originated from the study of erythropoiesis. In 1994, Moi et al. [9] identified NRF2 as one of the DNA-binding proteins that could bind to AP-1/NF-E2 (activating protein-1/nuclear factor erythroid 2) in the locus control region of β-globin, assisting NF-E2 to mediate globin gene expression. Subsequently, in 1997, pioneering work by Itoh and his colleague demonstrated that NRF2 acted an essential part in the transcriptional induction of phase II detoxification genes carrying ARE (antioxidant response elements), after which roles of NRF2 in erythrocyte generation were marginalized, with studies shifting their focus from erythropoiesis to toxicology [10].

Most importantly, so far, almost all genes that participate in ferroptosis are involved in the transcriptional regulation of NRF2, including glutathione regulation (such as System X_c_^−^ and GPX4), iron regulation (such as FTH1 (ferritin heavy chain 1), FTL (ferritin light chain) and FPN1 (ferrous iron exporter ferroportin 1)), NADPH regeneration (such as G6PD (glucose-6-phosphate dehydrogenase), and ME1 (malic enzyme 1)), and so on [11,12,13,14], which are described in this review too. In addition, studies have suggested that NRF2 indirectly participated in the regulation of lipid metabolism via PPARγ (peroxisome proliferator-activated receptor-γ) whose abundance was associated with cell sensitivity to ferroptosis [6]. Hence, with abundant upstream and downstream effectors, targeting NRF2 signaling pathway can effectively affect ferroptosis. In other words, therapies that promote or inhibit ferroptosis through modulating NRF2 signaling pathway are feasible and are worth the time and effort required for their exploration.

## 2. Ferroptosis

The key process of ferroptosis is Fenton reaction—Fe^2+^ and O_2_ (or ROS, reactive oxygen species) function as strong oxidants and peroxidize PL-PUFA (phospholipid with polyunsaturated fatty acyl tail) to PL-PUFA-OOH [15] in the catalysis of POR (cytochrome p450 oxidoreductases) [16] or LOXs (lipoxygenases) [17]. PL-PUFA-OOH then triggers the followed reactions of ferroptosis (Figure 1), though the mechanism of which is still not quite clear.

### 2.1. Processes That Promote Ferroptosis

Other than PE (phosphatidylethanolamines) [18], PUFAs (polyunsaturated fatty acids) might be the prime substrates for lipid peroxidation according to recent data [19]. PL-PUFAs are sensitive to peroxidation because of their C=C bond. PL-PUFAs are biosynthesized from acyl-CoA in multi-steps in the catalysis of ACC (acetyl coenzyme A carboxylase) [20,21], ACSL4 (acyl-CoA synthetase long-chain family member 4) [22], and LPCAT3 (lysophosphatidylcholine acyltransferase 3) [23]. PUFAs must be esterified into membrane PL-PUFAs so that they can be peroxidized to transmit the ferroptosis signal.

Energy stress (such as glucose starvation) inhibits lipogenesis and ferroptosis through activation of AMPK (adenosine-monophosphate-activated protein kinase), which inhibits the activity of ACC and biosynthesis of PUFAs [17]. E-cadherin-mediated cell–cell contacts activate Hippo signaling and thus inhibit nuclear translocation of NRF2 and the activity of transcriptional co-regulator YAP (Yes-associated protein), which promote transcription of ACSL4, TfR1 (transferrin receptor 1), and possibly other contributors of ferroptosis. Thus, closure of Hippo pathway and increased YAPs can make cells more sensitive to ferroptosis [24].

#### 2.1.1. LIP (Liable Iron Pool): Iron Drives Lipid Peroxidation with the Aid of ROS

Fe^3+^ formed from intestinal absorption or erythrocyte degradation can be captured by Tf and imported into cells through TfR1 [25], after which it is reduced to Fe^2+^ by STEAP3 (six transmembrane epithelial antigen of the prostate 3) [26] and imported via DMT1 (ferrous ion membrane transport protein) [27]. Fe^2+^ is maintained in the LIP, bound to low molecular weight compounds, including GSH (glutathione) [28].

The excess Fe^2+^ can be oxidized and then exported by FPN1 (ferrous iron exporter ferroportin 1) in the plasma membrane [29], utilized by mitochondria for heme synthesis, or oxidized and stored in ferritins by PCBP1 (poly-(rC)-binding protein 1) [30]. Heme synthesis in the mitochondria is regulated by ABCB6 (ATP-binding cassette subfamily B member 6) and FECH (ferrochelatase), which, respectively, transport coproporphyrinogen III and Fe^2+^ from the cytosol to the mitochondrial intermembrane space to form heme [31]. For heme degradation, HO-1 (heme-oxygenase 1) breaks down heme into biliverdin and liable Fe^2+^. Biliverdin is further metabolized to a potent antioxidant called bilirubin by BLVRA/BLVRB (biliverdin reductase A and biliverdin reductase B) [32,33]. Thus, HO-1 may play a dual role in ferroptosis. Some studies suggested that HO-1 promotes ferroptosis [34,35], while others reported opposing results that knockout or knockdown of HO-1 exacerbates ferroptosis [36,37,38].

Ferritins can be exported through MVBs (prom2-mediated ferritin-containing multivesicular bodies) when necessary [39], while some can be degraded into liable Fe^2+^ through ferritinophagy triggered by NCOA4 (nuclear receptor coactivator 4) [40]. In ferritinophagy, NCOA4 recruits ferritins into autophagosome, and then VAMP8 (vesicle associated membrane protein 8) mediates autophagosome–lysosome fusion [40], after which newly freed Fe^3+^ can be reduced and transported into cytosol LIP by STEAP3 and DMT1 [41]. Except NCOA4, there are other ferritinophagy activation proteins, such as ULK1 (unc-51 like kinase 1), atg5 (recombinant autophagy-related protein 5), and atg7 (recombinant autophagy-related protein 7) [42].

#### 2.1.2. Run-Out Mitochondrial Respiration Boosts ROS Generation

The out-of-control respiration of mitochondria is assumed to act as a promoting effect on ferroptosis because of the generation of more ROS [43]. Glutaminolysis is assumed to be an important factor for the run-out mitochondrial respiration. Glutamate is a precursor of α-KG (α-ketoglutaric acid), and hence, glutamate converted into α-KG accelerates the TCA (tricarboxylic acid) cycle and mitochondrial ROS generation (with hyperpolarization of inner mitochondrial membrane potential) [44].

### 2.2. Processes That Inhibit Ferroptosis

#### 2.2.1. Cyst(e)ine/GSH/GPX4 System

Cysteine and glutamate are essential for GSH biosynthesis to protect against oxidation. Cystine is imported by system X_c_^−^, reduced into cysteine by TXN/TXNRD1 or through transsulfuration pathway [45] (reductive power from NAD(P)H), and utilized in GSH synthesis in the catalysis of GCL (glutamate-cysteine ligase, or γ-GCS, γ-glutamylcysteine synthetase) and GSS (glutathione synthetase). GPX4 (glutathione peroxidase 4) converts GSH into GSSG (oxidized glutathione) and reduces L-OOH (lipid peroxides) to the corresponding L-OH (lipid alcohols) [46]. TXN (thioredoxin) and TXNRD1 (thioredoxin reductase) contribute to the reduction of cystine to cysteine [47,48].

System X_c_^−^ is an amino acid anti-porter in membrane, composed of two subunits, SLC7A11 (xCT) and SLC3A2, which imports cystine and exports glutamate in a ratio of 1:1 [4]. And SLC7A11 can be inhibited by p53 [49]. Since GPX4 is a selenoprotein, its biosynthesis relies on the co-translational incorporation of Sec (selenocysteine) [50].

#### 2.2.2. FSP1/CoQ_10_/NAD(P)H System

FSP1 (ferroptosis suppressor protein 1) can convert CoQ_10_ (ubiquinone) to CoQ_10_H_2_ (ubiquinol) with NAD(P)H, adding to the antioxidation capacity [51]. The intermediate of CoQ_10_, IPP (isopentenyl pyrophosphoric acid, or isopentenyl-PP), is as important as CoQ_10_ for the maturation of selenocysteine tRNA for GPX4 [52]. FSP1 can be activated by PPARα (peroxisome proliferator-activated receptor-α) [53].

#### 2.2.3. GCH1/BH_4_/DHFR System

GCH1 (GTP cyclohydrolase 1) is the rate-limiting enzyme for BH_4_ (tetrahydrobiopterin) synthesis. BH_4_ can reduce lipid peroxidants and be converted into BH_2_ (dihydrobiopterin), which can be regenerated from BH_2_ with NAD(P)H by DHFR (dihydrofolate reductase). BH_4_ can also promote the synthesis of CoQ_10_ by DHFR [54].

#### 2.2.4. Lipid Metabolism: PL-MUFA Inhibits Peroxidation of PL-PUFA

MUFA (monounsaturated fatty acid) biosynthesis requires SCD (stearoyl-CoA desaturase), which converts saturated fatty acids into MUFAs [55]. MUFA can be converted into MUFA-CoA by ACSL3 (acyl coenzyme A synthetase long-chain family member 3) and ultimately forms PL-MUFA (phospholipid with monounsaturated fatty acyl tail), the competing substrate against PL-PUFA [56]. Thus, the biosynthesis of PL-MUFA inhibits ferroptosis.

## 3. NRF2 Structure and Function

### 3.1. NRF2 Structure

NRF2 (nuclear factor erythroid 2-related factor 2, also named NFE2L2) contains 605 amino acids with seven highly conserved homology domains, named Neh1 (NRF2-ECH homology domain-1) to Neh7 (Figure 2). The Neh1 domain contains CNC/bZIP (Cap-N-Collar/basic leucine zipper) motifs, which can be bound with sMAF (small musculoaponeurotic fibrosarcoma) to form heterodimers that bind to AREs [57,58]. The amino terminal domain or carboxyl terminal domain is named Neh2 or Neh3, individually. The DLG and ETGE motifs on Neh2 domain bind with DC domain of Keap1 (kelch-like ECH-ssociated protein 1) homodimer [44,59], leading to the ubiquitination and degradation of NRF2 [60]. The Neh3 domain includes a VFLVPK motif, which is a critical binding site for CHD6 (chromodomain helicase DNA binding protein 6) [61,62]. Neh4 and Neh5 are acidic conserved sequences, which enhance NRF2 downstream genes expression via binding CBP (cAMP responsive element binding protein) [63,64]. Neh6, a serine-rich conserved sequence, could be phosphorylated by GSK-3β (glycogen synthase kinase 3β), leading to the degradation of NRF2 [63]. The Neh6 domain likely contains another phosphorylation-dependent motif, the DSAPGS motif, an extra binding site for β-TrCP (β-transducin repeat-containing protein) [65]. The Neh7 functions as a binding site for RXRα, capable of inhibiting NRF2 and its AREs [66,67].

### 3.2. NRF2 Regulation

The major approach to change NRF2 activity is regulating the stability of proteins, of which the Keap1-Cul3-Rbx1 axis is the most important regulator. In addition, NRF2 activity can also be regulated at the transcriptional and post-transcriptional levels.

As for the stability of proteins, Keap1 is considered to play a crucial role in the regulation of NRF2 stability. Therefore, NRF2 stability regulation is artificially divided into Keap1-dependent regulation and Keap1-independent regulation. As mentioned above, under normal circumstances, two molecules of Keap1 bind to the ETGE and DLG motifs on the Neh2 domain of NRF2 via its Kelch repeat domain [44,59]. As an adaptor protein for the Cul3 (Cullin3) E3 ubiquitin ligase complex, it can assemble with Cul3 and Rbx1 into a functional E3 ubiquitin ligase (Keap1-Cul3-E3), which leads to continuous ubiquitination and degradation of NRF2, ensuring low levels of intracellular NRF2 [59]. However, when cells are subjected to internal and external stimuli such as oxidative stress, Keap1 is inactivated due to modification of cysteine residues [68], competitive binding of p62 [69], and so on, causing NRF2 to dissociate from Keap1, and it becomes stable and subsequently transfers to the nucleus. Within the nucleus, NRF2 increases the transcription of antioxidant genes by interacting with other protein factors (such as sMaf) and binding to the ARE.

There are two main ways to regulate NRF2 stability independent of Keap1. One is the β-TrCP-Cul1-dependent pathway. Under the condition of homeostasis, NRF2 is phosphorylated by GSK-3β, after which it can be recognized by β-TrCP, resulting in the ubiquitination of NRF2 by the β-TrCP-Cul1 E3 ubiquitin ligase complex and later degradation by proteasome [63,70,71]. Under stress, the PI3K (phosphoinositide 3-kinase)/Akt pathway is activated, leading to the suppression of GSK-3β by phosphorylation, thus activating NRF2 [72]. The other is the Hrd1-dependent pathway. Hrd1 is an E3 ubiquitin ligase that binds directly to NRF2 and causes the ubiquitination and degradation of NRF2, negatively regulating NRF2 protein levels [73].

As for the regulation at the transcriptional level, Hrd1 is mainly involved in some transcriptional factors that can bind to specific sites in the promoter of NRF2 gene, such as AhR (aryl hydrocarbon receptor) [74], NF-κB (nuclear factor-κB) [75], NRF2 [76], and so on, which can increase the transcription of NRF2. Regulation of NRF2 also occurs at the post-transcriptional level. For example, NRF2 is regulated by a variety of miRNA genes, some of which (such as miR-144) can be locally complementary to the mRNA of NRF2 to silence the NRF2 gene [77]. In addition, in some tumor cells, the mRNA precursor of NRF2 can undergo alternative splicing, resulting in loss of the domain interacting with Keap1, so that Keap1 fails to bind to NRF2 normally, leading to NRF2 stabilization and a decrease in ferroptosis in cancer cells [78].

### 3.3. NRF2 Functions

#### 3.3.1. Antioxidant-Dependent Function

(1).Glutathione metabolism

Two crucial enzymes in GSH biosynthesis are regulated by NRF2—GCL (composed of GCLC/GCLM, glutamate-cysteine ligase catalytic/modulatory subunits) and GSS [11,76,79,80,81]. SLC7A11, one of the two subunits of system X_c_^−^, is also regulated by NRF2 in GSH metabolism, which is responsible for transporting cystine into the cell, thereby increasing cysteine content and promoting the process of GSH generation [80,81].

With the assistance of GSH, GPX4 exerts the activity of reducing peroxides. GSH serves as a reducing agent that donates electrons to xenobiotics or electrophilic organisms, thus transforming into its oxidized form GSSG. Research has shown that GPX4 mainly acts on H_2_O_2_ or organic hydrogen peroxide compounds, as well as ONOO^-^ [82]. The resulting agent, GSSG, is subsequently reduced for the maintenance of intracellular GSH pool. This process is completed by GSR (glutathione-disulfide reductase) with NADPH [83]. According to reports, both GPX4 and GR are found being transcriptionally regulated by NRF2 [84,85].

(2).FSP1-CoQ_10_ axis

FSP1 is a lipophilic antioxidant, which has been confirmed to be targeted by NRF2. The enzyme regenerates ubiquinol from ubiquinone, suppressing the formation of PL-PUFA-OOHs [19,86]. By regulating the expression level of FSP1, NRF2 maintains low levels of cellular peroxides.

(3).Components of the erythrotoxin family

Besides GSH, another system that resists oxidative stress is the erythrotoxin system. PRDX (peroxiredoxin) can reduce ROS level and is oxidized during this process. Oxidized PRDX is then reduced by TXN (thioredoxin), with the participation of cysteine oxidized by ROS. TXNRD1 (thioredoxin reductase 1) is used to rescue oxidized TXN with NADPH [87,88]. The expression of PRDX, TXN, and TXNRD1 mentioned above are all controlled by NRF2 at transcription level [89].

#### 3.3.2. Antioxidant-Independent Function

NRF2 can exert its antioxidant-independent effects by orchestrating the genes expression in phase II detoxification processes [90]. It activates the transcription of genes encoding detoxification enzymes such as GSTs (glutathione S-transferases) and NQO1 (NAD(P)H:quinone oxidoreductase 1) [90]. These enzymes play pivotal roles in cellular detoxification, facilitating the conjugation and elimination of harmful substances and xenobiotics. In the realm of inflammation, NRF2 activation demonstrates anti-inflammatory properties by modulating the expression of genes involved in inflammatory signaling pathways [91]. In recent years, NRF2 has been found to play a critical role in cellular metabolism.

(1).Heme/iron metabolism

As an E3 ubiquitin ligase, HERC2 is responsible for the degradation of NCOA4 (ferritin cargo receptor mediating ferritinophagy) and FBXL5 (IRP1/2 regulator mediating ferritin synthesis) and thus downregulates liable iron level. HERC2 is upregulated by NRF2 to fight against ferroptosis [40]. FTL and FTH1, light and heavy chains of ferritin, respectively, are controlled by NRF2. FTL is the main stabilizer of ferritin, and FTH1 contains active sites of iron oxidase, through which Fe^2+^ is oxidized into Fe^3+^ and is stored in the central core. FPN1/SLC40A1 responsible for iron excretion from cells is also controlled by NRF2. In addition, HMOX1/HO-1, ABCB6, FECH, SLC48A1 (member A1 of lipolytic vector family 48), and BLVRA/B are also positively regulated by NRF2 [31,32,92,93,94]. Among them, SLC48A1 is responsible for transporting the released heme from the lysosome to the cytoplasm after the degradation of hemoglobin in aged red blood cells, reducing the degradation of heme.

(2).NADPH generation in glucose metabolism

NADPH generation occurs in many pathways such as the PPP (pentose phosphate pathway), NADK (NAD kinase)-catalyzed phosphorylation of NADH, and the transformation of isocitrate to α-KG by IDH (NADP-dependent isocitrate dehydrogenase). NRF2 directly regulates the transcription of various PPP enzymes, such as G6PD (glucose-6-phosphate dehydrogenase) and other oxidative PPP enzymes [95], comprising reversible and irreversible reactions [96]. NRF2 transcriptionally regulates ME1 (malic enzyme 1) and IDH1 (isocitrate dehydrogenase 1) as well, contributing to the generation of NADPH [95]. NADPH is not only crucial for the synthesis of fatty acids (such as PUFAs), cholesterol, and nucleotides [97], but also important in antioxidation axes such as GPX4, FSP1, and DHFR [19,98,99].

(3).Lipid metabolism

Lipid synthesis highly consumes NADPH and competes with cellular antioxidant systems. Studies have shown that activation of NRF2 mitigates the generation and desaturation of fatty acid [13,54] to increase NADPH content for detoxification in murine liver.

PPARγ (peroxisome proliferator-activated receptor gamma) and NR0B2 (nuclear receptor subfamily 0, group B, member 2) are nuclear receptors that play significant parts in the regulation and control of cholesterol metabolism, which may aid in anti-peroxidation. Both genes can be activated by NRF2 and thus play a crucial role in mitigating accumulation of lipid peroxides [93,100,101].

(4).Amino acid metabolism

NRF2 positively regulates xCT (SLC7A11, the subunit of system X_c_^−^), which imports cystine and exports glutamate [102]. NRF2 also regulates TXN and TXNRD1 at transcriptional level, both of which help cystine to be reduced to cysteine [47,48]. SLC1A5 (solute-linked carrier family A1 member 5), which can be activated by NRF2, plays an essential role in glutamine uptake, which contributes to GSH biosynthesis [103]. But some studies showed that NRF2-actived cells are in a state of glutamate deficiency due to elevated xCT, GCLC/GCLM, and GSS. As a result, less α-KG is generated from glutamate and TCA cycle is restricted, hence inhibiting generation of ROS by ETC (electron transport chain), the downstream process of TCA cycle [104].

These antioxidant-independent functions of NRF2 mentioned above illustrate its multifaceted role in maintaining cellular homeostasis and influencing disease pathogenesis. Elucidating the intricate mechanisms underlying these functions holds promise for the development of innovative therapeutic approaches by targeting NRF2 activation in various pathological conditions associated with inflammation, metabolic disorders, and tissue damages.

## 4. NRF2 in Ferroptosis

NRF2 is kept low by three E3 ubiquitin ligases (KEAP1-CUL3-RBX1, SCF/βTrCP, and synoviolin/Hrd1). In the presence of gene mutation, endogenously induced modification, competitive binding of other interacting particles, and inhibition of exogenous drugs, NRF2 will be activated and transferred into the nucleus to initiate the transcription of target genes (Table 1).

### 4.1. Target NRF2 Signaling Pathway and Promote Ferroptosis in Cancer Therapy

Ferroptosis has emerged as a subject of significant interest within the cancer research field. Its distinct mechanistic and morphological attributes differentiate ferroptosis from other cell death modalities, thereby rendering it a promising avenue for cancer therapeutic exploration. The inhibition of ferroptosis observed across various cancer types, coupled with its dynamic role as a tumor suppressor during cancer progression, suggests that manipulating the regulation of ferroptosis holds promise as an interventional strategy in tumor treatment [105]. Consequently, small molecules that reprogram cancer cells to undergo ferroptosis are regarded as potent therapeutic agents for cancer therapy. Ferroptosis inducers can be classified into several different groups, each acting through different mechanisms.

#### 4.1.1. Four Types of FIN (Ferroptosis Inducers)

(1).Class I FIN: system X_c_^−^ inhibitors

System X_c_^−^ inhibitors, including erastin, can block uptake of cystine and reduce cysteine content, leading to a significant reduction in GSH levels, lipid ROS-mediated damage, and finally ferroptosis [46]. Biochemical and metabolomic analyses have demonstrated a notable decrease in GSH levels following erastin treatment [106]. Studies have also observed decreased GPX4 activity in various cancer cells treated with erastin [107]. Moreover, AUR (auranofin) is able to trigger ferroptosis in liver by inhibiting the activity of TXNRD and facilitating lipid peroxidation when given a high dose while a lower dose might be relatively safe [108]. Similar to erastin, SAS (sulfasalazine) can inhibit system X_C_^−^ and then trigger ferroptosis [109].

(2).Class II FIN: glutathione peroxidase 4 (GPX4) inhibitors

The conventional inducer of ferroptosis, RSL3 ((1S,3R)-RSL3), functions by covalently binding to the active site seleno-cysteine residue of the glutathione peroxidase 4 (GPX4) enzyme, irreversibly inhibiting its activity. This inhibition or loss of GPX4 activity results in the accumulation of lipid peroxides, which are considered lethal signals for ferroptosis. Different from RSL3, FIN56, derived from CIL56, can induce degradation of GPX4 with the participation of ACC and activate SQS which leads to CoQ_10_ depletion, rather than directly inhibit GPX4 activity [21]. Furthermore, exhaustion of GSH as well as the inactivation of GPX4 may play an important role in cisplatin-induced ferroptosis and apoptosis in A549 and HCT116 cells [110].

(3).Class III FIN: ubiquinone generation inhibitors

Statins, including fluvastatin, lovastatin, and simvastatin, are a class of pharmaceutical agents utilized for the reduction of blood cholesterol levels [111]. These drugs exert their cholesterol-lowering effects by inhibiting HMGCR (3-hydroxy-3-methylglutaryl-coenzyme A reductase), an enzyme responsible for the rate-limiting step in the mevalonate pathway, which is involved in cholesterol synthesis. By impeding the generation of isopentenyl pyrophosphate, an intermediate metabolite in the mevalonate pathway, statins can also hinder the biosynthesis of selenoproteins, including GPX4, and CoQ10, hence enhance ferroptosis [21]. The established clinical safety profile of statins, coupled with obesity being a significant risk factor for cancer, has led to the initiation of numerous clinical trials investigating the efficacy of statins as monotherapy or combination therapy in various types of tumors [112].

(4).Class IV FIN: lipid metabolism disruptors

Excessive iron levels induce the generation of reactive oxygen species (ROS), leading to an escalation in lipid peroxide levels. This lipid peroxide accumulation within cell membranes serves as a source of lipid ROS, which further contributes to the initiation of ferroptosis [2,3,4].

FINO2 (an endoperoxide containing 1,2-dioxolane) can directly oxidizes iron, leading to widespread lipid peroxidation via Fenton reaction and finally ferroptosis ultimately [113]. Lapatinib is able to attenuate the expression of ferroportin and ferritin and promotes the expression of transferrin, causing an increase in intracellular iron ion levels and subsequently ferroptosis [114]. Studies have shown that artemisinin derivatives induce ferroptosis in tumor cells by oxidizing iron, generating more ROS and fostering Fenton reaction [115].

#### 4.1.2. Clinical Application of Targeting NRF2 and Ferroptosis in Cancer Therapy

NRF2 can fight against ferroptosis by regulating its downstream antioxidant genes, but this also means that it maintains the survival of tumor cells while protecting normal cells. In the past decade, many studies have revealed that the activation of NRF2 in cancer cells promotes cancer progression [116,117,118] and metastasis [119], and confers cancer cells resistance to chemotherapy and radiotherapy [120,121]. Based on the mechanism of ferroptosis inducers mentioned above, drugs have been designed and invented in order to induce ferroptosis by targeting NRF2 signaling pathway in cancer therapy, and some of which are feasible enough to be utilized and evaluated in clinical trials (Table 2). Several drugs are described in detail below, by considering some cancer types as examples.

(1).Liver cancer

HCC (Hepatocellular carcinoma) is the primary malignant tumor in the liver, accounting for 90% of all liver tumors, and is the fourth leading cause of cancer death worldwide [141,142]. Sorafenib has been in the market for more than ten years as a first-line targeted drug for patients with advanced liver cancer. As a multi-kinase inhibitor, sorafenib targets tumors mainly by inhibiting tumor angiogenesis. Years of studies on the cytotoxicity of sorafenib in vitro have found that sorafenib induces an iron-dependent cell death different from apoptosis by inhibiting SLC7A11 [109], an element in the NRF2 downstream signaling pathway, thus blocking the transmembrane transport of cysteine and suppressing the biosynthesis of GSH. Results show that such death can be protected by inhibitors of ferroptosis such as liproxstatin-1, Fer-1 (ferrostatin-1) or DFO (deferoxamine), indicating that sorafenib can induce ferroptosis of liver cancer cells [143]. However, many cancer cell types are known to have resistance to ROS-induced chemotherapy drugs, partly via increasing the expression of some of key antioxidant genes. For instance, sorafenib resistance (both primary and acquired), in which upregulation of NRF2-related signaling pathway in cancer cells plays a core role, is considered to be the main cause of poor prognosis in HCC patients. Currently, some novel treatment strategies to overcome drug resistance has been developed [144]. For instance, the combination therapy of metformin and sorafenib on HCC cells can induce ferroptosis through the p62-Keap1-NRF2 antioxidant signaling pathway. By preventing the translocation of NRF2, metformin reduces HO-1 expression, making HCC cells more sensitive to sorafenib and enhancing its anti-tumor effect [122]. Artesunate can also be used to sensitize sorafenib-induced ferroptosis by promoting lysosomal function [145].

Some nutraceuticals composed of active chemicals derived from natural sources are also capable of combating HCC. For example, APG (Apigenin, 4′,5,7-trihydroxyflavone), a natural bioflavonoid extensively existing in numerous fruits and vegetables, can achieve anti-tumor effect by altering the miR-101/NRF2-related pathway [123]. As a post-transcriptional regulator, miRNAs (microRNAs), a type of noncoding RNA with the length of 18–25 nucleotides, target the 3′-UTRs of mRNA, inhibiting its translation or causing its degradation [146]. As for the pharmacological mechanisms of APG, it is able to upregulate the expression of miR-101 [123], which implies that APG can attenuate the expression of NRF2 at the protein and mRNA levels and reduce NRF2 target genes expression, thus inducing ferroptosis of tumor cells.

(2).Breast cancer

About 1.7 million people worldwide are diagnosed with breast cancer every year, and about 500,000 people die of breast cancer [147]. (1S,3R)-RSL3 can inhibit the peroxidase activity of GPX4 in BC (breast cancer) cells, causing the accumulation of peroxides and then ferroptosis [124]. It is worth mentioning that drug-resistant breast cancer cells often show stronger dependence on GPX4 and are more sensitive to GPX4 inhibitors [148]. Therefore, GPX4 can be a potential target to overcome drug resistance in breast cancer.

Fascin is able to enhance the vulnerability of breast cancer to erastin-induced ferroptosis by degrading xCT through ubiquitin-mediated proteasome degradation pathway [125]. In addition, siramesine, a lysosomotropic agent, and lapatinib, a potent double tyrosine kinase inhibitor of EGFR (epidermal growth factor receptor, ErbB-1) and ErbB-2 [114], can co-operatively induce ferroptosis-mediated cell death in BC cells [149,150]. Concerning the underlying mechanism, Ma et al. [150] demonstrated that the combination of siramesine and lapatinib attenuates the expression of ferroportin and ferritin and promotes the expression of transferrin, causing an increase in intracellular iron ion levels. Excess iron then participates in the Fenton reaction, raising the generation of ROS and leading to ferroptosis in MDA MB 231 and SKBR3 cells.

Moreover, LA (Levistilide A), an active compound extracted from Chuanxiong Rhizoma, is extensively utilized in cancer treatment in traditional oriental medicine. Previous studies have suggested that it is capable of promoting apoptosis in colon cancer cells through the ER (endoplasmic reticulum) stress pathway induced by ROS accumulation, exerting an anti-tumor effect [151]. In 2022, a study explored the underlying mechanism of LA-induced ferroptosis in vitro and eventually confirmed that LA treatment activated NRF2/HO-1 signaling pathway (upregulation of the expression of NRF2 and its downstream molecule HO-1), which enhanced ROS-induced ferroptosis in BC cells due to the dual effects of HO-1. Furthermore, LA also eliminate BC cells by significantly decreasing cell viability in a dose-dependent manner and disrupting the structure and function of mitochondria [131].

(3).Lung cancer

Lung cancer is responsible for massive cancer-related deaths worldwide. In 2018, the global incidence of lung cancer took the lead among all kinds of cancers [152]. NSCLC (non-small-cell lung cancer) is the most common type of lung cancer, representing over 80% of all lung cancers. DDP (Cisplatin) is regarded as the first-in-class drug for advanced and non-targetable NSCLC. Studies have indicated that DDP exerts the ability of inducing both ferroptosis and apoptosis in the NSCLC cell lines A549 and HCT116 cells [110]. It is confirmed that the major part of intracellular DDP can bind to GSH, forming the Pt-GS complex [153]. Therefore, similar to erastin, via GSH depletion and the inactivation of GPXs, DDP induces ferroptosis of cancer cells. However, it is essential to mention that anti-tumor effects of the combined application of erastin and DDP surpasses that of single use of DDP [110].

However, drug resistance to DDP is a major barrier to lung cancer treatment as well. A recent study revealed that the combination of isoorientin (also known as homoorientin) and DDP treatment resulted in an apparent decrease in the viability of drug-resistant cells through in vitro and in vivo experiments. In terms of mechanism, isoorientin acts as a modulator that controls the SIRT6/NRF2/GPX4 signaling pathway (downregulation of the expression of SIRT6 protein, NRF2, and GPX4), thereby regulating ferroptosis and reversing drug resistance in lung cancer cells [126].

A new drug, ZVI-NP (zero-valent-iron nanoparticle), which has been widely developed to treat contaminated groundwater or wastewater [154], is now gradually coming into sight for its anti-tumor efficacy. Previously, ZVI-NP has been found to be preferentially converted to ferric ions in lysosomes of cancer cells rather than in normal cells, since cancer cells are more acidic in intra-organelle compartments. Sudden release of iron ions further induced a ROS surge in cancer cells, and led to ferroptosis in cancer cells [155]. In 2021, when exploring the potential role of ZVI-NP in regulating the tumor microenvironment in lung cancer models, a study identified a novel molecular mechanism in inducing ferroptosis. Mechanistically, ZVI-NP enhanced phosphate-dependent ubiquitination and degradation of NRF2, thereby triggering ferroptosis along with excessive oxidative stress and lipid peroxidation. Moreover, ZVI-NP impaired the self-renewal capacity of cancer cells and inhibited angiogenesis in endothelial cells [127], all of which profoundly opened up the potential of new advanced cancer therapies that reduced side effects and enhanced efficacy.

(4).Pancreatic cancer

Pancreatic cancer ranks seventh in cancer-related mortality according to a survey conducted by the American Cancer Society in 2021 [156]. Moreover, a study has indicated that pancreatic carcinoma will represent the second leading cause of cancer-related deaths in the United States by 2030 [157]. Wogonin, as a main element composing *Scutellaria baicalensis* [158], is found to exhibit strong anti-inflammatory activity and potential for treating tumors in vitro and in vivo. A recent study has shown that wogonin decreases the GSH content in cells by suppressing the NRF2-mediated GPX4 pathway, leading to ROS accumulation and the enhancement of lipid peroxidation in pancreatic cancer cells [128], which indicates that wogonin could be potentially used for the treatment of pancreatic carcinoma.

(5).Other cancers

AS (Artesunate), a water-soluble derivative of artemisinin, induces ferroptosis in head and neck cancer cells by depleting GSH and causing ROS accumulation. Furthermore, silence of NRF2 by RNA interference or inhibition of NRF2 by trigonelline can enhance the killing effect of AS on drug-resistant head and neck cancer cells in vivo and in vitro [37]. Sulfasalazine is an azo-bridged agent with anti-inflammatory activity, originally synthesized from the antibiotic sulfapyridine in 1940. It was confirmed to be a potent inhibitor of xCT in 2001 [129]. Through disturbing cystine absorption, sulfasalazine mitigates the biosynthesis of GSH, resulting in ferroptosis [4] of some types of cancer cells in vivo and in vitro [130], such as lymphoma [129]. Sulfasalazine is supposed to be non-toxic so that it can be utilized in combination with other drugs to fight against drug resistance and side effects [159]. Okazaki et al. [160] found that the combination of dyclonine and sulfasalazine restrained the growth of head and neck squamous cell carcinoma or gastric tumors with high expression of ALDH3A1, which contributes to resistance to sulfasalazine monotherapy. Tagitinin C, a sesquiterpene lactone isolated from Tithonia diversifolia, is widely found in the Asteraceae [161], with ample pharmacological effects, including anti-tumor, anti-virus, anti-fibrosis, anti-parasite, and cardio-protective activity [162,163,164]. Mechanistically, it induced ER stress and oxidative stress, which activated the nuclear translocation of NRF2, resulting in oxidative cell microenvironment, ferroptosis, and suppressed growth of colorectal cancer cells. As a downstream effector of NRF2, the expression of HO-1 increased significantly, which led to an increase in the LIP and promoted lipid peroxidation. Furthermore, tagitinine C exhibited synergistic anti-tumor effects together with the erastin [132].

In general, it is reasonable to believe that more and more treatment strategies with higher efficacy and prognosis will be found in the future. The application of drugs that target NRF2 signaling pathway and induce ferroptosis to kill tumors is both novel and promising.

### 4.2. Targeting of NRF2 Signaling Pathway and Inhibition of Ferroptosis in Disease Therapy

#### 4.2.1. Diseases Associated with Ferroptosis

(1).Neurodegenerative diseases

Ferroptosis is presented in Alzheimer’s disease, Parkinson’s disease, and many other neurodegenerative diseases. Iron is the most abundant transition metal in the brain, participating in numerous metabolic reactions. Disorder of iron homeostasis in the brain can lead to some neurodegenerative diseases, such as Alzheimer’s disease and Parkinson’s disease [165,166,167]. Iron gradually accumulates in the brain with age, which changes the cerebral iron metabolism. In animal models of neurodegenerative diseases as well as aging populations, varying degrees of elevation of iron content in the brain have been observed [168,169]. Studies also show that chronic exposure to iron for mice causes dysfunction of membrane-transport protein, imbalance of intracellular iron homeostasis, and significant elevation of ROS and MDA (malondialdehyde), which ultimately lead to the dysfunction of nervous system [170]. Numerous studies reveal that disorder of iron metabolism has an influence on the normal folding of Aβ protein (amyloid β-protein) and phosphorylation of tau protein, resulting in oxidative stress and metal toxicity, which can cause DNA, lipid, and protein damage and ultimately lead to Alzheimer’s disease [171,172,173]. Studies also indicate that cognitive function of Alzheimer’s patients can be stabilized when applying antioxidants and iron chelators like α-lipoic acid, which blocks tau-induced iron overload, lipid peroxidation, and ferroptosis-related inflammation [174]. Parkinson disease is characterized by mitochondrial dysfunction, iron accumulation, copper depletion, and GSH exhaustion in the brain, some of which are also the most important features of ferroptosis [175,176,177,178,179]. Iron chelators (ferroptosis inhibitor) like deferiprone show therapeutic effect in Parkinson’s patients in a randomized controlled trial [180].

(2).Cardiovascular diseases

Ferroptosis plays an influential role in cardiovascular diseases. Recent studies have found a significant decrease in GPX4 expression in the early and middle stages of myocardial infarction [181]. Human umbilical cord blood-derived mesenchymal stem cell exosomes may inhibit ferroptosis by targeting miR-23a-3p, thereby achieving a protective effect in alleviating acute myocardial infarction [182]. Ferroptosis also plays an important role in hypertension. Pulmonary hypertension can cause a decrease in GSH levels and an increase in iron, ROS, and lipid peroxides [183]. The cardiotoxicity of anthracyclines (e.g., doxorubicin) is also associated with ferroptosis. Studies have shown that doxorubicin can lead to heme degradation, causing mitochondrial dysfunction and a significant increase in the content of iron, ROS, and other markers of ferroptosis [35]. In DOX (doxorubicin)-induced cardiomyopathy, the administration of ferroptosis inhibitor Fer-1 (ferrostatin-1) can significantly reduce the mortality rate of mice, while the mortality rate of mice administered with inhibitors of apoptosis, necrosis, and autophagy is not significantly reduced [35]. Ferroptosis may also play a significant role in sepsis. In animal models of sepsis with cecum ligation and puncture, the levels of GSH and GPX4 decreased accompanied by elevated levels of iron and lipid peroxides [184]. Many other types of cardiovascular diseases, such as arrhythmias and sickle cell anemia, are also considered to be associated with ferroptosis [185,186].

(3).Organic injuries

Ferroptosis has been investigated in many organic injuries as well. For example, in mice with sepsis, multiple organic injuries caused by ferroptosis can be detected. Ferroptosis has also been proved to play an important role in LPS-induced acute lung injury, in which Fer-1 also displays effective therapeutic effect [187]. In various acute renal injuries, like rhabdomyolysis-induced acute kidney injury and folic acid-induced acute kidney injury, iron accumulation and ROS have been observed and can be suppressed by ferroptosis inhibitors [188,189,190]. Fe^2+^ produced by myoglobin metabolism directly induces lipid peroxidation in proximal tubular epithelial cells in rhabdomyolysis, which may be an important mechanism of rhabdomyolysis inducing acute kidney injury [189]. And the application of iron chelator deferoxamine is able to alleviate rhabdomyolysis-induced kidney injury in mice [191].

#### 4.2.2. Drugs and Clinical Experiments

There have been many studies focusing on the role of NRF2 and showing that regulation of NRF2 and related signaling pathway can alleviate neurodegeneration, cardiovascular diseases, and many organic injuries.

(1).Neurodegenerative diseases

NRF2 has promising prospects as a novel target for the treatment of neurodegenerative diseases, since Alzheimer’s disease is closely related to oxidative stress, mitochondrial dysfunction, and ferroptosis. The activation of NRF2 can upregulate the expression of xCT, enhance glutamate secretion, and restore GPX4 activity, thereby inhibiting ferroptosis. Therefore, drugs activating NRF2 could become a new approach to treat Alzheimer’s disease. DMF (Tecfidera^®^) can be regarded as the most successful case reported in activating NRF2 [192,193]. The drug is originally used to treat psoriasis patients, but new research shows that after entering the body, it transformed so as to covalently modify key cysteine residues on KEAP1, thus blocking NRF2 ubiquitination and promoting NRF2 stabilization as well as subsequent activation of NRF2 target genes [133]. In addition, many drugs that can be used to treat Alzheimer’s disease by activating NRF2, such as resveratrol, pyridoxine, and NBP (n-butylphthalide), have also entered clinical trials.

(2).Cardiovascular diseases

Targeting NRF2 has broad prospects in the treatment of cardiovascular diseases. SIRT1 (Sirtuin1) is a NAD-dependent deacetylase which plays an important role in a variety of cardiovascular diseases caused by diabetes. It can increase the localization of NRF2 in the nucleus, thereby alleviating myocardial ischemia-reperfusion injury in the diabetic Sprague Dawley rat model [134]. The effect of SIRT1 may be achieved by deacetylation of NRF2 and reduction of NRF2 ubiquitination [194]. At the same time, miRNAs can indirectly regulate NRF2 expression by modulating the upstream protein NRF2. MiR-24-3p that has been shown to modulate the Keap1/NRF2 pathway during myocardial ischemia-reperfusion [135]. In highly glucose-stimulated rat and mouse cardiomyocyte models, inhibition of miR-144, miR-155, and miR-503 activates NRF2 to attenuate cellular oxidative stress [195,196,197]. MiR-24 has been shown to activate the NRF2/HO-1 signaling pathway, playing a critical role in combating oxidative stress in VSMCs (vascular smooth muscle cells) stimulated by high glucose [198].

(3).Organic injuries

In brain injury related to transient global cerebral ischemia in rats, study shows that preischemia dexmedetomidine administration has neuroprotective effect by enhancing NRF2/HO-1 expression and reducing caspase-3 activity [136]. In ischemia brain injury, BCP (β-Caryophyllene) has protective effects, whose mechanism is related to the activation of the NRF2/HO-1 pathway [137]. Moreover, study shows that XJZ (Xiaojianzhong decoction), which consists of six Chinese herbal medicines and extracts, can attenuate ferroptosis mediated by oxidative stress in the gastric mucosal injury by upregulating the p62/Keap1/NRF2 pathway [138]. In sepsis-induced lung injury, AS (artesunate) upregulates the expression of antiferroptosis systems like NRF2 and HO-1, thus attenuating neutrophil infiltration and pathological damage in lung tissue [139]. Studies in the type II diabetic nephropathy rat model show that experimental diabetic nephric injury can be alleviated by targeting the Keap1/NRF2 pathway using MiR-200a [140].

Although drugs targeting NRF2 show therapeutic functions, and have a promising future, studies on these drugs are still limited to the experimental stage, and further clinical trials are needed for application. Multiple aspects, including safety, efficacy, and economy, should be taken into consideration before clinical application.

## 5. Conclusions and Future Perspectives

With a further understanding of the mechanism of ferroptosis, the criticality of NRF2 is demonstrated more obviously, whose rich value embodies in both antioxidant-dependent and antioxidant-independent functions. As a key regulator of the cellular antioxidant response, NRF2 is widely involved in iron, lipid, amino acid, glucose metabolisms, etc. Therefore, through pharmacological modulation of NRF2 and its downstream effectors, prevention and treatment of pathologies, such as numerous types of cancers, neurodegenerative diseases, cardiovascular diseases, and so on, can be achieved. Despite several unclear issues, such as the specific connection between lipid peroxidation and ferroptosis, there is no denying the fact that therapies targeting NRF2 signaling pathway exert extraordinary efficacy. This kind of approach can be used to treat various ferroptosis-related diseases and may have a brilliant therapeutic future.

## Figures and Tables

**Figure 1 antioxidants-12-01739-f001:**
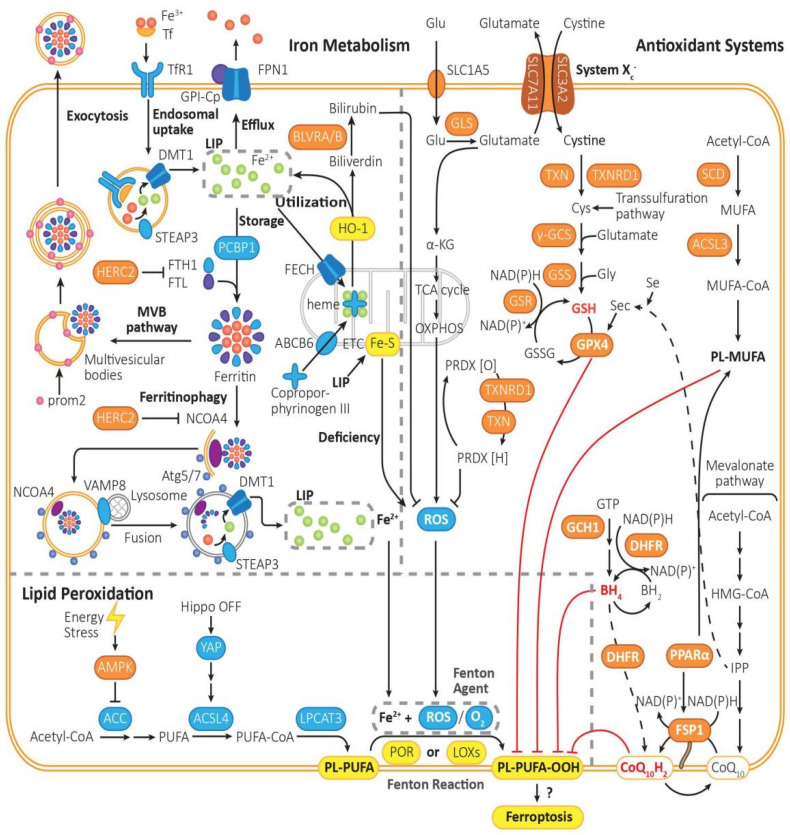
The mechanism of ferroptosis. Ferroptosis is driven by iron-dependent lipid peroxidation, the reaction of PL-PUFA with Fe^2+^ and ROS under the catalysis of POR or LOXs. Ferroptosis is regulated by lipid metabolism, iron metabolism, and intracellular antioxidant system. (1) Lipid metabolism: PL-PUFA, the prime substrate for lipid peroxidation, is synthesized from Acetyl-CoA in multiple steps, which are catalyzed by ACC, ACSL4, and LPCAT3. Then, it is prone to be peroxidized to PL-PUFA-OOH, which triggers the followed reactions of ferroptosis, though the initiation mechanism is still not quite clear. The peroxidation process is mainly driven by Fenton reaction, or mediated by POR or LOXs. By activating AMPK, energy stress can inhibit the activity of ACC, thus turning on an energy stress-protective program against ferroptosis. (2) Iron metabolism: The peroxidation of PL-PUFA is driven by the labile iron pool and iron-dependent enzymes. Fe^3+^ is imported into cells by Tf through TfR1, then being reduced by STEAP3 to Fe^2+^, forming the labile iron pool. Excess Fe^2+^ can be expelled via FPN1, be stored in ferritin, which can be exported through MVB pathway or release Fe^2+^ through ferritinophagy, or be utilized to synthesize heme with coproporphyrinogen III through ABCB6 and FECH, or regenerated when heme is metabolized by HO-1. (3) Intracellular antioxidant system: There are mainly three pathways for removing PL-PUFA-OOHs. Cyst(e)ine/GSH/GPX4 axis: It is one of the most important pathways in ferroptosis inhibition. It requires uptake of cystine via system X_c_^−^, reduction of cystine to cysteine, and biosynthesis of GSH. GPX4-mediated reduction of PL-PUFA-OOH to PL-PUFA-OH, and regeneration of GSH from GSSG. FSP1/CoQ10 axis: It functions independently of GPX4. CoQ_10_H_2_ can remove lipid peroxides, and then consume NADPH to regenerate CoQ_10_H_2_ by FSP1. GCH1/BH_4_/DHFR system: BH_4_ can be synthesized from GTP under the catalysis of GCH1, being converted into BH_2_ and clearing lipid peroxidants, and then regenerated by DHFR. Furthermore, saturated fatty acid can be transformed into PL-MUFA in multi-steps, which is the competing substrate against PL-PUFA and can suppress ferroptosis. Furthermore, PRDX can reduce the content of ROS, which can be generated from PRDX[O] through NADPH and TRX.

**Figure 2 antioxidants-12-01739-f002:**
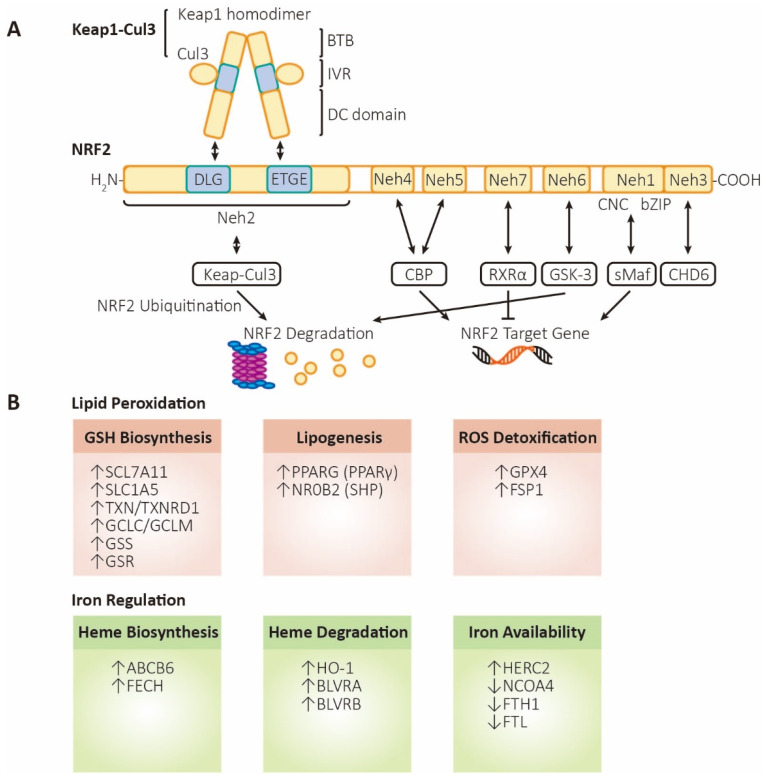
The structure and functions of NRF2. (**A**) Structure and regulation of NRF2. Keap1, kelch-like ECH associated protein 1; Cul3, cullin3 E3 ubiquitin ligase; BTB, broad complex, tramtrack, and bric-a-brac domain; IVR, intermediate region; DC domain, DGR (double glycine repeat) and CTR (carboxyl terminal region) domain; NRF2, nuclear factor erythroid 2-related factor 2; Neh1~6, NRF2-ECH homology domain-1~6; CNC, Cap-N-Collar motif; bZIP, basic leucine zipper motif; sMaf, small musculoaponeurotic fibrosarcoma; CHD6, chromodomain helicase DNA binding protein 6; CBP, cAMP responsive element binding protein; GSK-3, glycogen synthase kinase 3; RXRα, retinoid X receptor alpha. (**B**) Functions of NRF2. Lipid peroxidation (labeled in orange, referring to processes that inhibit ferroptosis): SLC7A11 (xCT), solute carrier family 7 member 11, a heteromeric, sodium-independent, highly specific cysteine-glutamate transport system; SLC1A5, solute carrier family 1 member 5, a sodium-dependent neutral amino acid transporter; TXN, thioredoxin; TXNRD1, thioredoxin reductase 1; GCLC/GCLM, glutamate-cysteine ligase catalytic/modifier subunit, comprising heterodimer γ-GCS (gamma-glutamylcysteine synthetase); GSS, glutathione synthetase; GSR, glutathione-disulfide reductase; PPARG (PPARγ), peroxisome proliferator-activated receptor gamma; NR0B2 (SHP), nuclear receptor subfamily 0 group B member 2 (small heterodimer partner); GPX4, glutathione peroxidase 4; FSP1 (AIFM2), ferroptosis-suppressor-protein 1 (apoptosis inducing factor mitochondria associated 2). Iron regulation (labeled in green, referring to processes that promote ferroptosis): ABCB6, ATP-binding cassette subfamily B member 6, a member of the heavy metal importer subfamily, playing a role in porphyrin transport; FECH, ferrochelatase; HO-1, heme oxygenase 1; BLVRA/BLVRB, biliverdin reductase A/B; HERC2, HECT, and RLD domain containing E3 ubiquitin protein ligase 2; NCOA4, nuclear receptor coactivator 4, androgen receptor coactivator; FTH1, ferritin heavy chain 1; FTL, ferritin light chain.

**Table 1 antioxidants-12-01739-t001:** Major roles of NRF2 in ferroptosis (via regulating downstream genes).

Symbol	Whole Name	Function	Upregulation or Downregulation	Main Factors for Inhibiting Ferroptosis
ABCB6	ATP-binding cassette subfamily B member 6	Heme synthesis (transport protoporphyrin to mitochondria)	↑upregulation	Decrease the capacity of LIP
FECH	Ferrochelatase	Heme synthesis (insert ferrous iron into protoporphyrin)	↑upregulation	Decrease the capacity of LIP
FTH1	Ferritin heavy chain 1	Iron storage (oxidize ferrous iron into ferric iron)	↓downregulation	Decrease the capacity of LIP
FTL	Ferritin light chain	Iron storage (stabilize ferritin)	↓downregulation	Decrease the capacity of LIP
FPN1	Ferroportin 1	Iron excretion	↑upregulation	Decrease the capacity of LIP
HO-1	Heme-oxygenase 1	Heme degradation (decompose heme into ferrous iron and biliverdin)	↑upregulation	Increase the capacity of LIP but generate antioxidants (HO-1 exerts dual effects on ferroptosis)
BLVRA/B	Biliverdin reductase A/B	Heme degradation (metabolize biliverdin into bilirubin)	↑upregulation	Generate antioxidant: bilirubin
HERC2	HECT and RLD domain containing E3 ubiquitin protein ligase 2	Inhibit ferritinophagy (NCOA4 and FBXL5 ubiquitination and degradation)	↑upregulation	Decrease the capacity of LIP
SLC7A11	Solute carrier family 7 member 11	GSH synthesis (import cystine and export glutamate)	↑upregulation	Generate antioxidant: GSH
SLC1A5	Solute carrier family 1 member 5	GSH synthesis (absorb glutamine)	↑upregulation	Generate antioxidant: GSH
TXN/TXNRD1	Thioredoxin/Thioredoxin reductase 1	GSH synthesis (reduce cystine to cysteine)	↑upregulation	Generate antioxidant: GSH
GCLC/GCLM	Glutamate-cysteine ligase catalytic/modifier subunits	GSH synthesis (connect glutamate and cysteine)	↑upregulation	Generate antioxidant: GSH
GSS	Glutathione synthetase	GSH synthesis (connect glycine and γ-glutamyl cysteine)	↑upregulation	Generate antioxidant: GSH
GSR	Glutathione-disulfide reductase	GSH synthesis (reduce GSSG to GSH)	↑upregulation	Generate antioxidant: GSH
GPX4	Glutathione peroxidase 4	ROS detoxification (reduce lipid peroxides by oxidizing GSH to GSSG)	↑upregulation	Decrease the accumulation of lipid peroxides
FSP1	Ferroptosis-suppressor-protein 1	Ubiquinol synthesis (convert ubiquinone to ubiquinol with NAD(P)H)	↑upregulation	Generate antioxidant: ubiquinol
PPARγ	Peroxisome proliferator-activated receptor gamma	ROS detoxification (mitigate lipid peroxidation as a nuclear receptor)	↑upregulation	Decrease the accumulation of lipid peroxides
NR0B2	Nuclear receptor subfamily 0, group B, member 2	ROS detoxification (mitigate lipid peroxidation as a nuclear receptor)	↑upregulation	Decrease the accumulation of lipid peroxides

**Table 2 antioxidants-12-01739-t002:** NRF2 inhibitors and activators.

Inhibitors				
Compound	Model Organism	Mechanism	Diseases	References
Sorafenib	HT-1080 Calu-1 et al.	Inhibit NRF2/xCT signaling pathway	HCC	[109]
Metformin	HepG2, HUH-7, BALB/C nude mice	Inhibit NRF2 nuclear translocation	HCC resistant to sorafinib	[122]
APG	BEL-7402,BEL-7402/ADM	Enhance NRF2 degradation (upregulate miR-101 expression)	HCC	[123]
(1S,3R)-RSL3	BJeLR et al.	Inhibit NRF2/GPX4 signaling pathway	BC	[124]
Fascin	HS578T MDA-MB-231 et al.	Inhibit NRF2/xCT signaling pathway (induce xCT degradation)	BC	[125]
Isoorientin	A549	Inhibit SIRT6/NRF2/GPX4 signaling pathway	Lung cancer resistant to DDP	[126]
ZVI-NP	H1299, H460, A549	Enhance NRF2 degradation (by GSK3/β-TrCP)	Lung cancer	[127]
Wogonin	AsPC-1, PANC-1, HPDE6-C7	Inhibit NRF2/GPX4 signaling pathway	Pancreatic cancer	[128]
Trigonelline	Athymic BALB/C male nude mice	Silence NRF2 by RNA interference	Drug-resistant head and neck cancer	[37]
Sulfasalazine	OSC19, HSC-2 et al.	Inhibit NRF2/xCT signaling pathway	Lymphoma, head and neck squamous cell carcinoma, gastric tumor	[129,130]
**Activators**				
**Compound**	**Model Organism**	**Mechanism**	**Diseases**	**References**
Levistilide A	MDA-MB-231, MCF-7	Activate NRF2/HO-1 signaling pathway	BC	[131]
Tagitinin C	CRC cell lines	Enhance NRF2 nuclear translocation	Colorectal cancer	[132]
DMF	Mouse embryonic fibroblasts (MEFs)	Enhance NRF2 stabilization (modify key cysteine residues on Keap1)	Alzheimer’s disease	[133]
SIRT1	Diabetic Sprague Dawley rat model, H9C2	Enhance NRF2 nuclear translocation	Myocardial ischemia-reperfusion injury	[134]
MiR-24-3p	C57BL/6 mice, H9C2	Inhibit Keap1 expression Activate NRF2/HO-1 signaling pathway	Myocardial ischemia-reperfusion injury	[135]
MiR-24	Diabetic Sprague Dawley rat model VSMCs	Activate NRF2/HO-1 signaling pathway	Diabetics (oxidative stress in VSMCs stimulated by high glucose)	[135]
Dexmedetomidine	Transient global cerebral ischemia rat model	Activate NRF2/HO-1 signaling pathway	Brain injury related to transient global cerebral ischemia	[136]
BCP	Cerebral ischemia/reperfusion SD rat model	Enhance NRF2 nuclear translocation	Ischemia brain injury	[137]
XJZ	C57BL/6 mice	Activate p62/Keap1/NRF2 pathway	Gastric mucosal injury	[138]
AS	Specific-pathogen-free (SPF) male Kunming (KM) mice	Enhance NRF2 nuclear translocation	Sepsis-induced lung injury	[139]
MiR-200a	Goto-Kakizaki (GK) rat model	Enhance NRF2 nuclear translocation (inhibit Keap1 translation)	Experimental diabetic nephric injury	[140]

## Data Availability

Not applicable.

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
