# Peer review of "NRF2, a Superstar of Ferroptosis"

_antioxidants, 2023, doi:10.3390/antiox12091739_

Round 1
Reviewer 1 Report
This is an interesting review on the role of NRF2 in the ferroptosis, a recent identified programmed cells death mechanism that has attracted much attention. The work is well-organized and written, with some English errors.
- Abstract. The sentences at lines 13-15 should be improved. “In cancer cells, ferroptosis is frequently found abnormal suppression.” Something is missing! “While during tissue damages, ferroptosis is recurrently promoted, resulting in a large number of cell deaths and ultimately loss of the functions of the corresponding organs.” Loss of function of the organ seems an exaggeration.
- Line 52: “could bound” correct could bind
L. 236 “NFR2 are emerging” correct NFR2 is emerging
- The description of NFR2 structure is adequate, but its regulation is missing. A description of its effectors, promoter, transcriptional and post-transcriptional regulation would improve the work.
- The two figures are well-done and complete, but too complex to be attractive. A table that summarizes the major roles of NFR2 in ferroptosis would facilitate the reading.
I indicated the sentences that should be improved
Author Response
Responses to Reviewer 1
Thank you for your constructive and kind suggestions.
Our responses to your comments:
This is an interesting review on the role of NRF2 in the ferroptosis, a recent identified programmed cells death mechanism that has attracted much attention. The work is well-organized and written, with some English errors.
- The sentences at lines 13-15 should be improved. “In cancer cells, ferroptosis is frequently found abnormal suppression.” Something is missing! “While during tissue damages, ferroptosis is recurrently promoted, resulting in a large number of cell deaths and ultimately loss of the functions of the corresponding organs.” Loss of function of the organ seems an exaggeration.
Response: Thanks. We did the revision as you suggested.
- Line 52: “could bound” correct could bind.
Response: Thank you. We did the revision as you suggested.
- 236 “NFR2 are emerging” correct NFR2 is emerging.
Response: Thank you. We did the revision as you suggested.
- The description of NFR2 structure is adequate, but its regulation is missing. A description of its effectors, promoter, transcriptional and post-transcriptional regulation would improve the work. The two figures are well-done and complete, but too complex to be attractive. A table that summarizes the major roles of NFR2 in ferroptosis would facilitate the reading.
Response: Thank. As you suggested, we summarized the major role of NRF2 in ferroptosis in Table1 in our revised version. Besides, we simplify Figure 1 to make it more attractive.
- Comments on the Quality of English Language
I indicated the sentences that should be improved.
Response: Thank you. As you suggested, we improved our English in our revised version.

Reviewer 2 Report
In this manuscript Yan et al, review the mechanisms of ferroptosis and its regulation by NRF2.
The article is interesting and well written. The references are appropriated. The topic is relevant and deals with both therapeutic aspects of ferroptosis: Activation for cancer treatment and inhibition for neurodegenerative diseases. The main suggestion for improving the manuscript is to summarize the information in tables.
- Line 100. Mitochondrial iron is also utilized for Fe-S cluster biosynthesis. Please, include this information in Figure 1.
- Tables summarizing NFR2 inhibitors and activators, specifying model organism, disease, outcome, and references would be helpful for the readers.
Author Response
Responses to Reviewer 2
Thank you for your nice suggestions.
Our responses to your comments:
In this manuscript Yan et al, review the mechanisms of ferroptosis and its regulation by NRF2.
- The article is interesting and well written. The references are appropriated. The topic is relevant and deals with both therapeutic aspects of ferroptosis: Activation for cancer treatment and inhibition for neurodegenerative diseases. The main suggestion for improving the manuscript is to summarize the information in tables.
Response: Thank you for your nice advice. As you suggested, we summarized the role of NRF2 in ferroptosis in Table 1 in our revised version.
- Line 100. Mitochondrial iron is also utilized for Fe-S cluster biosynthesis. Please, include this information in Figure 1.
Response: Thank you. As you suggested, we included the information you mentioned in Figure 1.
- Tables summarizing NFR2 inhibitors and activators, specifying model organism, disease, outcome, and references would be helpful for the readers.
Response: Thank you for your nice advice. As you suggested, we summarized NRF2 inhibitors and activators in Table 1 in our revised version.
Round 2
Reviewer 2 Report
The authors have addressed all my concerns